# SailBuoy Ocean Currents: Low-Cost Upper-Layer Ocean Current Measurements

**DOI:** 10.3390/s22155553

**Published:** 2022-07-25

**Authors:** Nellie Wullenweber, Lars R. Hole, Peygham Ghaffari, Inger Graves, Harald Tholo, Lionel Camus

**Affiliations:** 1Plentzia Marine Station (PiE-UPV/EHU), University of the Basque Country, 48620 Plentzia, Spain; nellie.wullenweber@gmail.com; 2National Oceanography Center, University of Southampton, Southampton SO17 1BJ, UK; 3Faculty of Sciences, University of Liège, 4000 Liège, Belgium; 4Norwegian Meteorological Institute, 5007 Bergen, Norway; 5Akvaplan-Niva, 0579 Oslo, Norway; pgh@akvaplan.niva.no (P.G.); lca@akvaplan.niva.no (L.C.); 6Aanderaa-Xylem, 5225 Nesttun, Norway; inger.graves@xylem.com (I.G.); harald.tholo@xylem.com (H.T.)

**Keywords:** SailBuoy, ADCP, ocean current, observation

## Abstract

This study introduces an alternative to the existing methods for measuring ocean currents based on a recently developed technology. The SailBuoy is an unmanned surface vehicle powered by wind and solar panels that can navigate autonomously to predefined waypoints and record velocity profiles using an integrated downward-looking acoustic Doppler current profiler (ADCP). Data collected on two validation campaigns show a satisfactory correlation between the SailBuoy current records and traditional observation techniques such as bottom-mounted and moored current profilers and moored single-point current meter. While the highest correlations were found in tidal signals, strong current, and calm weather conditions, low current speeds and varying high wave and wind conditions reduced correlation considerably. Filtering out some events with the high sea surface roughness associated with high wind and wave conditions may increase the SailBuoy ADCP listening quality and lead to better correlations. Not yet resolved is a systematic offset between the measurements obtained by the SailBuoy and the reference instruments of ±0.03 m/s. Possible reasons are discussed to be the differences between instruments (various products) as well as changes in background noise levels due to environmental conditions.

## 1. Introduction

Offshore upper-ocean current measurements are time consuming and expensive and the coverage of in situ measurements is sparse. Numerical models and remote sensing observations are dependent on verification with in situ data. Industries affiliated with the ocean as well as research institutes are interested in well-modeled upper-layer ocean currents, as currents affect offshore infrastructure, shipping, oil spills, or the capability of modeling the trajectory of missing persons or boats at sea. In coastal areas, the knowledge of currents is increasingly important when considering beach erosion, transport of suspended matter, forces acting on marine structures, and navigational safety. Since models do require in situ data for improvement and verification, ongoing research, therefore, is trying to reduce the costs and resources of current measurements in order to close the gap between little available, robust, and accurate data and a high demand. Small unmanned autonomous vehicles have started complementing or even replacing traditional ocean observations as they require less power, ship time, and crew and provide an environmentally friendly and efficient alternative to the traditional methods [1]. It is, however, not straightforward to transfer traditional ocean measurement techniques to new platforms. For the measurement of ocean currents, one probable solution has been introduced by Offshore Sensing AS and Aanderaa Data Instruments AS who have equipped a remotely operated surface vehicle powered by wind and solar energy with an acoustic doppler current profiler (ADCP) in order to measure upper-layer ocean currents: the SailBuoy Ocean Currents. Measuring upper-layer ocean currents from an unmanned surface vehicle (USV) could be a promising alternative that overcomes the drawbacks of stationary moorings or the needs of a large research vessel. By deploying an ADCP on a USV, costs, deployment, and retrieval time are kept to a minimum while large geographic areas can be covered.

The SailBuoy (SB) is propelled solely by wind and designed for long-term autonomous operations for oceanographic and meteorological research objectives inshore and offshore. Its navigation and endurance capabilities have been widely tested, showing, even in the rough environmental conditions of the North Sea and the Barents Sea, a high resilience against the wind, waves, and extreme temperatures (e.g., [2,3,4]). The SB can be equipped with different sensors, while the necessary energy for navigation and instrumentation is provided by solar panels as well as batteries, allowing it to be autonomously at sea for multiple consecutive months.

The SB has been designed for easy deployment and recovery and has been kept simple and light to prevent it from getting caught up in debris or avoid damage to existing infrastructure at sea, e.g., moorings. To commercial ships, the SB with its comparably small size offers only little risk.

Similar approaches have been tried before, e.g., on the C-Enduro [5] or the Saildrone (https://www.saildrone.com/news/saildrone-sensor-suite-antarctica-adcp, accessed on 18 June 2022) for which validation studies are in the process of being published [6]. However, compared to these (e.g., the Saildrone has a length of 7 m and weighs up to 500 kg), the SB is relatively small and light, making it easier to handle, deploy, and retrieve at lower costs. Smaller USVs equipped with ADCPs have been deployed in estuarian or riverine environments but to our knowledge not in rougher offshore environments [7].

The SB itself is of ca. 2 m length and 1.13 m height, with a weight of 60 kg and space for 60 L of payload with a maximal weight of 15 kg. It behaves much like a sailboat traversing and tacking to make its way to user-defined waypoints. Depending on the wind conditions it can travel at speeds of up to 1 m/s [8,9]. The SB uses the Iridium satellite system (https://www.iridium.com/solutions/autonomous-systems/, accessed on 18 June 2022) for two-way communication, able to send its GPS location as well as receive commands and waypoints for remote operation. Its track and summary data can be viewed and controlled in real time via a website. Due to the high modularity of the vessel’s technology, a multitude of sensors (depending on size and weight) can be integrated into the vessel allowing next to water temperature, salinity, and dissolved oxygen concentration, also the monitoring of wave height (heave), algae, crude oil, wind or air pressure. Prior to the *SailBuoy Ocean Currents* project, the SB has already proven to be of scientific relevance on multiple occasions, e.g., to measure near-surface temperatures, salinity, and oxygen concentrations around Gran Canary Island and in the Gulf of Mexico [3,4], and for wave measurements in the North Sea [2,10]. Other scientific, as well as industrial applications, are imaginable, such as marine mammal monitoring, emission monitoring, oil spill tracking, visual inspections, a communication relay station for subsea instrumentation, or echosounder surveys for seismic monitoring.

Advantages of the SB compared to traditional measurement techniques are

Easy and low-cost deployment, retrieval, and maintenance;High modularity and flexibility;Coverage of the energy demand by solar panels;Real-time data transmission;Autonomous coverage of large geographical areas.

A downward-looking ADCP (600 kHz DCPS 5400) manufactured by Aanderaa Data Instruments AS has been integrated into the SB Ocean Currents. The maximum measurable depth range is 80 m depending on the settings and the scattering conditions of the water. Since the SB is susceptible to wave and wind conditions, a careful postprocessing of the data is required in order to calculate the absolute velocity of the water column, in the view of the degrees of freedom of the vessel housing, particularly the tilt.

The main objective of this study was to compare and validate the SB ADCP measurements against stationary current observations. Two validation campaigns took place in very different environments, typifying Norwegian fjord and offshore systems (see Figure 1). (1) A stationary experiment in Fusafjorden, Norway (October 2020) representing a secured and controlled inshore system; (2) an offshore deployment near the oil- and gasfield Ekofisk in the Norwegian sector of the central North Sea (March–April 2021). The first validation campaign was set up as a stationary experiment without the sail mounted on the vessel in order to create a secure measurement environment and ensure continuous valid data over time. The second experiment then took place offshore with the sailing and self-navigating SB to test its capability of providing data while under sail and making way over ground. During each campaign, there was a minimum of two reference instruments that the data of the SB could be compared to. See instrument specifications listed in Table 1.

## 2. Materials and Methods

### 2.1. Instrumental Setup–Stationary Fjord Deployment

The first experiment for obtaining validation data for the SB Ocean Currents was conducted in October 2020 in Fusafjorden, Norway (see map in Figure 1). As a branch of the Bjørnafjorden, Fusafjorden is a 13 km long fjord in Vestland county, ca. 30 km south of Bergen, which separates part of the Bergen peninsula from the mainland. It is a wide fjord that further to the north-east splits into three smaller branches. The water depth is ca. 42 m at the study location. The experiment aimed to measure the velocity field simultaneously from multiple instruments using the following mooring setup. A bottom-mounted upward-looking ADCP at ca. 42 m depth, a single-point acoustic current meter placed at almost mid-depth (ca. 20 m), and finally, the downward-looking SB ADCP fastened to the surface buoy of the mooring (see Figure 2). Specifications and setting of the instruments are provided in Table 1. The cell setting for bottom-mounted and SB ADCPs were 30 cells (2 m cell thickness) and 40 cells (1 m cell thickness), covering a range of 3–30 m, and ca. 40 m to the surface, respectively, while having a 50% overlap of the individual cells. The SB ADCP sampling frequency was 4 cph, where data were stored every 15 min based on a 10-min measurement period. The sampling frequency for both bottom-mounted ADCP and the single-point current meter was 6 cph. Hence, the datasets were interpolated to the SB ADCP sampling frequency for further processing. The mean pressure records were used for aligning cells depth and all units were unified. The further processing of the data is discussed more thoroughly below.

### 2.2. Instrumental Setup–Offshore Deployment Ekofisk

The second experiment took place offshore with the sailing and self-navigating SB to examine its capability of providing data while under sail and making way overground. In close cooperation with the Ekofisk operation management, the SB maneuvered along two given transect lines as close to the reference ADCPs as possible without getting into conjunction with marine traffic or logistic operations around the oil platforms. Data of four reference instruments in the area were available. Three upward-looking current profilers, moored at 42 m and 49 m, respectively, and the third installed on the Eldfisk platform ca. 16 km south of Ekofisk (see map in Figure 1). Additionally, a single-point current meter at 10 m depth was deployed on the Ekofisk Lima (Eko-L) platform.

A correlation analysis was performed on spatial subsampling to estimate the maximum distance the SB ADCP could be away from the bottom-mounted ADCPs, where it would still measure similar circulation features. On that account, the number of data points had to be balanced against the spatial variability (scale of eddies/currents in the region). Additionally, ancillary data sets such as significant wave height and wind that could influence the quality of current measurements were collected by different devices at the Eko-L platform.

### 2.3. Data Analysis

For a proper comparison and robust validation, the datasets were carefully scrutinized and matched temporally and spatially. Unifying data formats was also another challenge since seven different instruments from two manufacturers were used (see Table 1). While this allowed for a robust validation, it increased the workload of reading and formatting the data. The applied data preparation and preprocessing methods before comparing the signals of the different instruments are highlighted below.


*Horizontal and vertical alignment*


Spatial overlapping in the horizontal domain was ensured during the stationary deployment of the SB in Fusafjorden. However, the SB location had to be matched with the validation instruments during nonstationary deployment, i.e., while the SB was sailing autonomously. For the comparability in the vertical dimension, a correct localization of the collected data vertically in the water column was essential (bin sizes and cell depths). For point meters, this could be done simply via the pressure sensor. For ADCPs, however, this meant also the vertical resolution, i.e., the cell size and cell overlap of the instrument had to be taken into account. In this case and if not mentioned otherwise, a vertical interpolation was carried out in order to match the depth cells of the different profilers.


*Temporal interpolation*


To match the different time intervals of the datasets, e.g., for the correlation or frequency analysis, the datasets were brought onto the same temporal resolution via linear interpolation. In most cases, a temporal resolution of 15 min was chosen corresponding to the SB sampling interval. The results of the linear interpolation were assessed carefully.


*Filtering*


The measured datasets were filtered according to the detected backscatter intensity and standard deviations. In general, however, filtering did not have much influence on the results. Most of the datasets were surprisingly good without the need for taking out faulty data. Merely the start and end of measurement periods were cut by a few data points to exclude data spikes during deployment and retrieval periods. The SB ADCP data were also fairly good, with just a couple of occasions of extreme high values that were multiple times higher than the standard deviation. This was treated by truncating entirely or interpolating (for the frequency analysis). In some cases for the upward-looking bottom-mounted or moored ADCPs, surface interference had to be truncated, with a threshold according to the instrument providers’ manual [11].


*Pressure and tidal influence*


Pressure data of bottom-mounted instruments revealed a 1 m tidal amplitude range, which could affect the mapping of the depth cells and eventually lead to mismatching the depth cells of stationary and moving ADCPs. However, due to the cell thickness setup that provided a good overlapping and weighted average, the influence of the tidal amplitude was assumed to be relatively minor. Additionally, the vertical interpolation of the velocity field was another way to overcome this issue.


*Frequency analysis*


The frequency analysis was done by applying a fast Fourier transform (FFT) on the complex current velocity vectors to separate tidal and residual frequencies. For the tidal filter, we used the major semidiurnal and diurnal tidal component for the regions in question (see, e.g., [12]).

To check the FFT results, a harmonic analysis with the aid of the python version of T_TIDE (*ttide_py*, available on GitHub: https://github.com/moflaher/ttide_py, accessed on 18 June 2022; Matlab original version by Rich Pawlowicz [13]) was additionally done.

## 3. Results

### 3.1. Stationary Deployment in Fusafjorden

Figure 3 shows the velocity field observed by the bottom-mounted and SailBuoy ADCPs. Both observations reveal more or less similar flow field patterns. However, there are some slight differences between those observations. We note that the bottom-mounted ADCP shows somewhat higher current speeds and more damped current direction changes, which warranted a further data analysis.

The time series of current speed and direction at ca. 20 m depth measured by the single-point meter (middle) as well as the SB ADCP (top) and bottom-mounted ADCP (bottom) data are presented in Figure 4. Here, again, measurements of different instruments show a similar trend; however, the bottom-mounted ADCP recorded higher speeds compared to the others while also showing a larger standard deviation. The correlation coefficients between the SB ADCP and bottom-mounted and single-point current meter velocity measurements at a 20 m depth were 0.40 and 0.76, respectively.

Histogram graphs (Figure 5) also reveal the systematic offset between the bottom-mounted ADCP and the rest of the observations. While the observations of the SB ADCP and single-point current meter almost entirely overlap, the speed values recorded by the bottom-mounted are detached from the rest with a mean deviation of 0.018 m/s. It should be noted here that the current profiler on the SB and the single-point current meter are both from the same manufacturer (Aanderaa Data Instruments AS), while the bottom-mounted profiler is a Nortek instrument. The bias here directly opposes the one found during the Ekofisk deployment where the SB measured higher values than the Nortek bottom-mounted ADCP as is later discussed.

Figure 6 displays the time-averaged current speed calculated for each depth cell as well as the maximum measured speed at each depth measured by the SB ADCP and the reference instruments in Fusafjorden. The time-averaged vertical structure of the flow field, i.e., higher current speeds closer to the surface and decreasing with depth, is almost similar for both current profilers. However, a systematic velocity offset of approximately 0.02 m/s is also evident along the water column. The average velocity of the single-point current meter nicely fits with the SB records (deviating only by 0.001 m/s, which is less than the accuracy range of the instrument). Furthermore, the maximal measured velocity corresponds quite well between both instruments (deviating by 0.016 m/s). Even though the bottom-mounted profilers’ records are higher, the SB ADCP and the single-point current meter records are still within the range of the standard deviation.

A correlation analysis of the measurements over depth resulted in relatively high correlation coefficient values of more than 0.6 in the upper part of the water column (at 5–12 m) and values mostly below 0.4 from 15 m downward (see Figure 7). While the SB data correlated well with the Aanderaa point meter at 20 m depth (correlation coefficient 0.75), the bottom-mounted instrument exhibited a comparably poor correlation with the Aanderaa point meter (0.36). The *p*-values for all correlations done here vanished to zero, indicating that the positive correlation between the instruments, especially in the surface layers, was significant.

### 3.2. Offshore Deployment at Ekofisk

The active measurement period of the SailBuoy ADCP, while it was in proximity to the ADCPs (within 2–41 km range, see Figure 8), was almost one month. A further analysis was carried out in the view of the homogeneous current conditions in the area, which was inferred from the diverse models (e.g., European North West Shelf model, https://www.copernicus.eu/en/access-data/copernicus-services-catalogue/atlantic-european-north-west-shelf-ocean-physics-analysis, accessed on 18 June 2022) and the bottom-mounted ADCP records. A thorough data analysis, literature research (e.g., [14]), as well as weighing the quantity of data against possible uncertainties due to small eddy features, eventually led to the assumption of current conditions sufficiently homogeneous for the purpose of validation in the area of interest.

The current speed and direction time series (Figure 9) showed a good agreement between the SB and reference ADCPs.

Figure 10 presents the correlation analysis for all four profilers as well as the current point meter at the Ekofisk Lima (Eko-L) platform. The depth profiles of the time-averaged current speeds of all profilers are all fairly consistent and quite homogeneous, which contrasts with the results from the fjord deployment. The results reveal a slight offset (higher values) in the SB ADCP measurements. Furthermore, the correlations are higher among the moored and bottom-mounted instruments. The average values of the SB records, however, are within the standard deviations of the reference instruments, with a deviation of approximately 0.03 m/s from the measurements of ADCP 1 and 2, which is in contrast to the results obtained from the inshore deployment in Fusafjorden and requires a further investigation (see Figure 6). A possible influencing factor that should not be neglected here, is the difference in the flow field in the protected fjord and offshore regions. In general, the recorded velocities in Fusafjorden were approx. 50 % lower than those at the Ekofisk site.

Apart from the average values, the maximum current speeds are also approximately constant with depth for ADCP1, 2, and the SB. Only the Eldfisk measurements show an increase towards the upper cells. This might be due to a strong high current event in early April that affected the Eldfisk measurements and distorted its maximum measured speeds towards the upper layers. It is also noteworthy that the results underline the assumption of an approximately barotropic and horizontally homogeneous flow field structure throughout the area of interest during most of the measurement period.

The current speed recorded by the SB correlates fairly well over the entire depth range, i.e., above 0.5, with the reference profilers (see the right panel in Figure 10). In particular, higher correlations were achieved between the SB and ADCP2, which were around 0.65 for below 20 m depth and dropped slightly (r = 0.55) in the upper layers. Here, the average correlation for the entire water column was 0.63. This observation opposed the one during the stationary fjord deployment where the correlation was higher for the upper part of the water column. The correlation analysis was significant since the *p*-values for all depths vanished towards zero (similar to the Fusafjorden experiment), which was well below the significance level of 0.05.

The correlation analysis between the SB and ADCP1 velocity records exhibited a different correlation profile compared to the other ADCPs with higher correlation coefficients in the upper cells and lower in the lower layers, similar to the Fusafjorden experiment. The average correlation coefficient was 0.54, which was somewhat smaller than the one for the SB and ADCP2. Here, it is important to highlight that the SB ADCP and ADCP1 records were only overlapped from ca. 16 to 31.5 m, and the cell thickness setting was different (see Table 1). Additionally, the SB was geographically closer to the ADCP2 than ADCP1 during most of the operation time.

The correlation with the Eldfisk data is also represented in Figure 10. Here, the overlapping depth range is even smaller and the upper cell values of the Eldfisk ADCP might be distorted due to a high current event in early April. However, still, for depths below ca. 22 m, the correlation with the SB is above 0.5 even though Eldfisk is 20 km south of the Ekofisk platforms. This again could be evaluated in the view of the homogeneous current conditions, especially at depth.

A correlation analysis was also applied to radius-restricted SB datasets, i.e., only including data when the SB was within a 2, 5, and 10 km distance from the reference instruments (not shown). Results indicated no significant changes in the correlation coefficients for the depth profiles or the depth-averaged time series. Furthermore, the correlation even increased when increasing the radius, i.e., including data points from greater distances. This is due to the strong tidal signals and reinforces our assumption of a barotropic circulation pattern allowing for a continuous and homogeneous flowfield in the study area.

For reference a correlation analysis was also performed between the various reference instruments resulting in unsurprisingly high values of 0.70 for ADCP1 and ADCP2 and values around 0.57 and 0.46 for the correlation between Eldfisk and ADCP2 and ADCP1, respectively, (not shown).

Of interest now are those observation periods associated with weaker correlations, to understand which circumstances negatively affect the SB observation skills. In that connection, ancillary data of environmental conditions such as wave height, wind speed, and the direction from the Ekofisk Lima platform were the most relevant parameters for further investigation. The upper two panels in Figure 11 show the depth-averaged absolute current speed of the SB, ADCP1, and ADCP2 and the deviation between the SB measurements and those of the two profilers. Data from the Eldfisk ADCP were left out here for a better visualization. The observation results show that the storm event around 6 April had a large impact on all datasets. It is of great advantage that this event happened during the campaign since it allowed the estimation of the SB’s measurement performance in extreme weather conditions, with waves up to a height of 8 m and wind speeds of up to 20 m/s. The event lasted ca. 5 days from 5 to 10 April. The extreme conditions are reflected in the SB’s motion pattern.

The frequency of the changes in the vessel’s speed and its tilt was much higher during this extreme weather period and tilting angles of up to 90° were more frequent. Comparing the ancillary environmental data with the depth-averaged current speeds confirmed the hypothesis that strong weather events can lead to a perturbation in the SB measurement capabilities. The large deviations between the datasets during the event also led to the assumption that a higher tilting or higher frequencies of changes in the SB speed may hamper the processing software tool’s capability to calculate the correct current speed from the backscattered signals. It was also evident that the higher cruise speeds of the SB did not decrease the data quality. Deviations were low during periods in which the SB traveled relatively fast with up to 0.4 m/s, as seen for example during the first days as well as the last days of the campaign (around 1 April and again after 25 April). It appeared that it was not the SB speed itself but the frequency at which it changes and its tilt that influenced the accuracy of the measurements.

Apart from the large storm event around 6 April, two other heavy weather conditions are noteworthy: the high wind speeds and waves around 13 April and again around 22 April. Both events were of shorter duration and measured less extreme magnitudes of wave height and wind speeds compared to the one of 6 April, but led again to high tilting angles of the SB and higher frequencies in the changes of the vessel’s speed over ground. In both events, the influencing factors were very similar. The measurements of the SB deviated from the reference instruments during the second event on 20–23 April, but not during the first event (10–14 April). The only detectable difference and therefore assumed main reason for this finding was that the current speed during the second event was significantly lower compared to the first event (ca. 0.05 vs. 0.15 m/s). Low currents are generally harder to detect since signal-to-noise ratios can be higher, leading to higher errors in the data.

A signal decomposition using a fast Fourier transform (FFT) and a harmonic analysis showed the SB skill in capturing the velocities associated with the main tidal component. On average, the correlations between the tidal signals of the SB and reference ADCPs were above 0.76 (not shown). Even in the previously discussed rough weather conditions the SB measured current velocity comparably well, except for the heavy weather event in early April, where the signals were distorted even at tidal frequencies.

Filtering the SB data for calm conditions, i.e., taking out data points measured during high waves (e.g., 3 m) or SB tilt angles of 50°, resulted in an increase in correlation coefficients from 0.59 (0.69) to 0.75 (0.81) for ADCP1 and ADCP2, respectively, (see Figure 12). This confirmed the prior found results of heavy wave conditions and hence the movement of the SB influencing its measurement capabilities. It also indicated that a simple postprocessing of the data could lead to highly satisfying results.

## 4. Discussion

### 4.1. Stationary Deployment in Fusafjorden

The field survey results showed that current velocities in the Fusafjorden were almost 50% lower than at the offshore sites. The semiprotected fjord condition probably leads to a stratified flow field and weaker periodic signals. Despite this, the SB measured similar current profiles as the bottom-mounted ADCP. The correlation coefficients were found to be highest in the surface layers with a maximum of 0.73 at around 10 m, but decreasing to below 0.3 for the deeper layers. A simple explanation could be the vertical velocity shear, i.e., slightly higher velocities towards the upper layers. The maximum velocity curves seemed to covary with the correlation coefficient profiles. The observations suggested that higher current speeds could increase the measurement skill of both instruments, or at least the detection of the same predominant features.

This first validation campaign allowed for the comparison of comparably shallow measurement cells of the SB (up to ca. 5.5 m) which yielded correlation coefficients above 0.65. Slightly disrupting the confidence in the data, however, was the systematic negative offset of ca. −0.02 m/s between the current speed measurements by the SB and bottom-mounted ADCP. The fact that this offset had the opposite sign compared to the results found during the second validation campaign at Ekofisk where the SB was also compared to reference profilers is puzzling. Unfortunately, there were no wind and wave sensors deployed on the surface mooring during the Fusafjorden campaign. Such data could have been used for a similar comparison as that done for the Ekofisk site to see if this could be a possible factor leading to the bias. From the available data, however, the underlying reason for the offset can only be hypothesized. If assuming that the reference profiler is the one that measures accurately, the only possible explanation for the bias are slow current conditions. The fact that the single point meter at 20 m depth measured very similar values to the SB is contradicting this hypothesis. Hence, the bias here is not provoked by inaccurate measurements obtained by the implementation of an ADCP on the SB but by varying measurement capabilities of the instruments of different providers. Nortek has produced ADCPs for many years and their profilers are state-of-the-art instruments. Current point meters, however, in general are accepted to deliver more accurate results as they are point measurements. Especially the Seaguard RCM by Aanderaa has correlated in a variety of studies and always performed very well [15]. It could therefore be reasonable to assume that the measurements of the SB ADCP—since it measures the same as the point meter—are more precise. This again could be a result of the lower frequency employed in the bottom-mounted Nortek profiler at 400 kHz vs. the 600 kHz Aanderaa profiler on the SB. A lower frequency results in a larger single pin standard deviation. This is expanded upon in the following when discussing the Ekofisk dataset.

### 4.2. Offshore Deployment at Ekofisk

The results of the offshore experiment showed a good correlation between the SB measurements and traditional moored profilers. However, a comparison among the datasets also revealed a positive offset of approx. 0.03 m/s, which necessitated a closer look. The instrument providers of the moored Aquadopp profilers (Nortek) and the Aanderaa DCPS onboard of the SB give accuracy values for the instruments of 0.005 m/s (or ±1% of the reading, Nortek) and 0.003 m/s (or ±1.5%, Aanderaa). Applying an accuracy range of 1.5% to the values measured here in the order of ca. 0.1 m/s results in a possible error range of ±0.0015 m/s, therefore not including the offset’s magnitude measured by the SB. When trying to find the ultimate reason for the bias in both measurements, one could try to point out the differences between the offshore and inshore deployment. These are, however, numerous, starting from the stationary, i.e., not maneuvering SB without a sail, to in general slower currents and lower tidal action in the inshore environment. Another possible influencing factor could have been the changed setting of the SB ADCP during the Ekofisk deployment, where one instead of five records (as done in Fusafjorden) per 15 min was collected. Each record having 150 pings resulting in 150 vs. 750 pings per sampling interval. This could have an influence on the signal-to-noise ratio leading to higher noise values in the SB measurements at Ekofisk. The fact that the bias was reduced by over 30% when filtering out high-frequency noise suggests that the magnitude of the offset is strongly influenced by the amount of noise in the measurements. This could point to the conclusion that the SB measurements are more strongly affected by noise (e.g., due to wave action), especially because the bias is lower (and even negative) in the fjord environment where wave and wind conditions are assumed to have been weaker. Unfortunately, different settings of the individual profilers such as sampling interval, average interval, and ping rate make a direct comparison infeasible, so that a final conclusion on the offset can not be given here.

The offset being opposite to the one in Fusafjorden contradicts the earlier hypothesis that merely different instrument providers could have led to different measurements. However, if one still follows this chain of thought, could it be that the Aanderaa instruments measure in general higher values in high current conditions and lower in low current conditions, while Nortek profilers measure lower values in high current conditions but higher in low currents? This question goes beyond the scope of this work and must be tested in a different experiment. If, on the other hand, the point meters at both sites were taken as the only accurate reference measurements, this would indicate that the SB measured correctly in presumably calmer wave and wind conditions inshore and detected too high values offshore in rougher weather conditions. This would confirm the hypothesis that the measurements from the SB as a less stable ADCP platform are more strongly affected by motion due to wind and waves. However, this again requires further investigation.

Another explanation for the opposite sign of the bias is the following: the 400 kHz bottom-mounted Nortek profiler used in the Fusafjord has a larger inherent single ping standard deviation due its the lower frequency compared to the 600 kHz profiler installed on the SB. The increased noise was seen in Figure 4. This increased noise level compared to the single point current meter and the SB ADCP could be an explanation for the negative bias in the Fusafjord campaign. In the Ekofisk trial, the SB ADCP current measurement standard deviation was increased due to a lower ping number per measurement interval, and an increased measurement noise due to the increased wave situation. This could in turn be the explanation for the positive bias of the SB ADCP vs. the bottom-mounted ADCP in the Ekofisk trial. Furthermore, the correlation analyses applied to the datasets of the reference instruments resulting in correlation coefficients of similar magnitude as those of the SB ADCP data revealed the difficulty of obtaining accurate current measurements for validation.

Apart from the bias, thanks to the observed storm events (with waves up to a height of 8 m and wind speeds of up to 20 m/s), the SB’s measurement capability could be evaluated in different environmental conditions while safely maneuvering around the Ekofisk oil and gas platforms. A comparison of the measurements during the storm events on 6 April, 13 April, and 21 April resulted in a list of factors that were shown to influence the correlation results. These included but were not limited to the wave conditions, wind speed, and possibly its direction, the current speed, as well as the tilt and speed of the SB itself.

It became clear that the SB detected current velocities especially well in high current conditions during periods of little and low frequent changes in the SB’s movement. Furthermore, it remains a challenge to measure current velocities in low current conditions with high-frequency changes in motion of the autonomous vessel itself. The motion due to changes in wind speed or wave height adds noise to the data leading to a decrease in the signal-to-noise ratio. However, it was shown that even then, the high-frequency noise could be filtered out with standard tools of frequency analysis (e.g., FFT or harmonic analysis and tidal prediction) providing good estimates of the tidal currents in the measurements. This was true in waves up to 3 m and winds up to 13 m/s, but could not be done for long periods of extreme storm conditions. Filtering the measurements of the SB according to its tilt and wave height can therefore help improve the accuracy as was shown. Of advantage for this as well as for a possible inclusion into a metocean measurement campaign is the SB’s high modularity, which would allow for additionally fitting wind and wave sensors onto the vessel. This, however, always must be balanced out for questions of power supply and endurance. This ancillary data could be used for model validation as well as for data quality screening of the SB current data itself. The advantages of having these additional sensors and disadvantages of a higher energy consumption must be balanced, well planned, and calculated before each campaign. Before a possible inclusion of the SB in upcoming measurement campaigns, the slight deviations during moderate to heavy weather events as well as the detected offset must be considered and eventually resolved depending on the required accuracy.

## 5. Conclusions

The two field experiments discussed here, represented very different environmental conditions, allowing for an in-depth data analysis and confidence in the results. A comparison with traditional current measurement devices showed that the ADCP by Aanderaa Data Instruments AS on board the SB can deliver satisfying results. However, two opposing systematic offsets in the measurements (+0.03 m/s offshore and −0.02 m/s inshore) have yet to be fully explained before the SB Ocean Currents can be involved in metocean campaigns. Possible reasons were suggested to be:The difference in the absolute current speed in both campaigns (almost 50% lower in Fusafjorden than at the Ekofisk site).Higher wave and wind conditions adding noise to the Ekofisk data.The changed setting to fewer records and hence number of pings for each sampling interval (one record at Ekofisk instead of five as in Fusafjorden)Differences in the instruments by the different providers (Nortek/Aanderaa), especially the operating frequencies, leading to deviations in the detection of slower/stronger current speeds.

The factors influencing the measurement capabilities of the SB were furthermore found to be the wave and wind conditions, the strength of the current, and the frequency of the movement of the SB. The best measurement capabilities were found in conditions of high current speeds and relatively low wave and wind action. For future applicability of the SB, it was therefore proposed to include sensors for wave and wind measurements into the vessel. Here of course, the payload size and power supply, especially for longer deployment periods, need to be considered.

Apart from this, the SB proved to be able to withstand heavy weather conditions and to safely maneuver back to a specified harbor for recovery. The deployment was observed to be simple and easily applicable on a research cruise, which makes it interesting for any metocean observation program. The Ekofisk deployment also showed that with thorough planning and communication, the SB can even be securely deployed in areas of crucial offshore operations and high, frequent marine traffic. Furthermore, the SB enables current measurements of surface layers which are usually—especially by shipborne ADCPs—not measurable since the instruments are located further down the ships’ hull or experience interference from the surface in the case of bottom-mounted/moored ADCPs.

Finally, with the results presented here, especially the simple handling during deployment and recovery, and under the prerequisite that the issue of the offset is resolved, the SB Ocean Currents has the capability of becoming a low-cost alternative to traditional current measurement devices and deliver data for metocean observation programs and model validation.

## Figures and Tables

**Figure 1 sensors-22-05553-f001:**
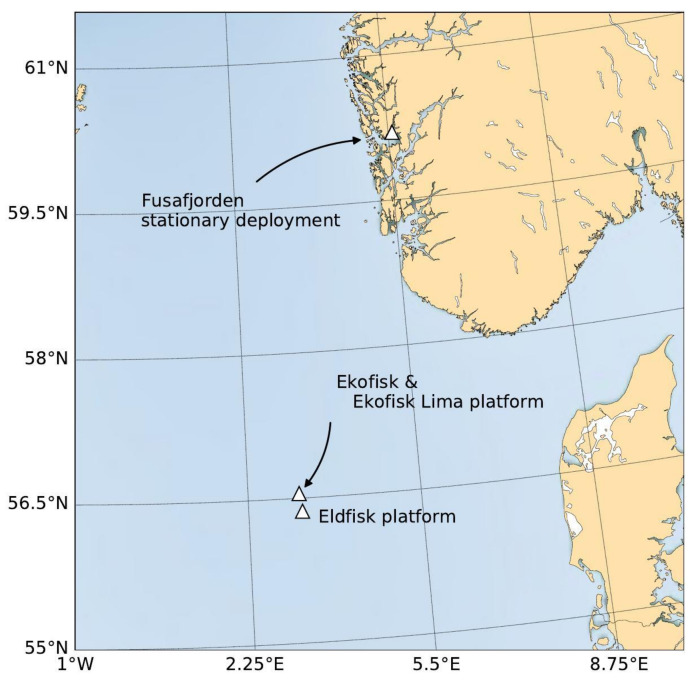
Map of the central North Sea with marked locations where the validation campaigns took place: the stationary deployment in Fusafjorden and the undersail deployment near the Ekofisk and Eldfisk oil and gas platforms.

**Figure 2 sensors-22-05553-f002:**
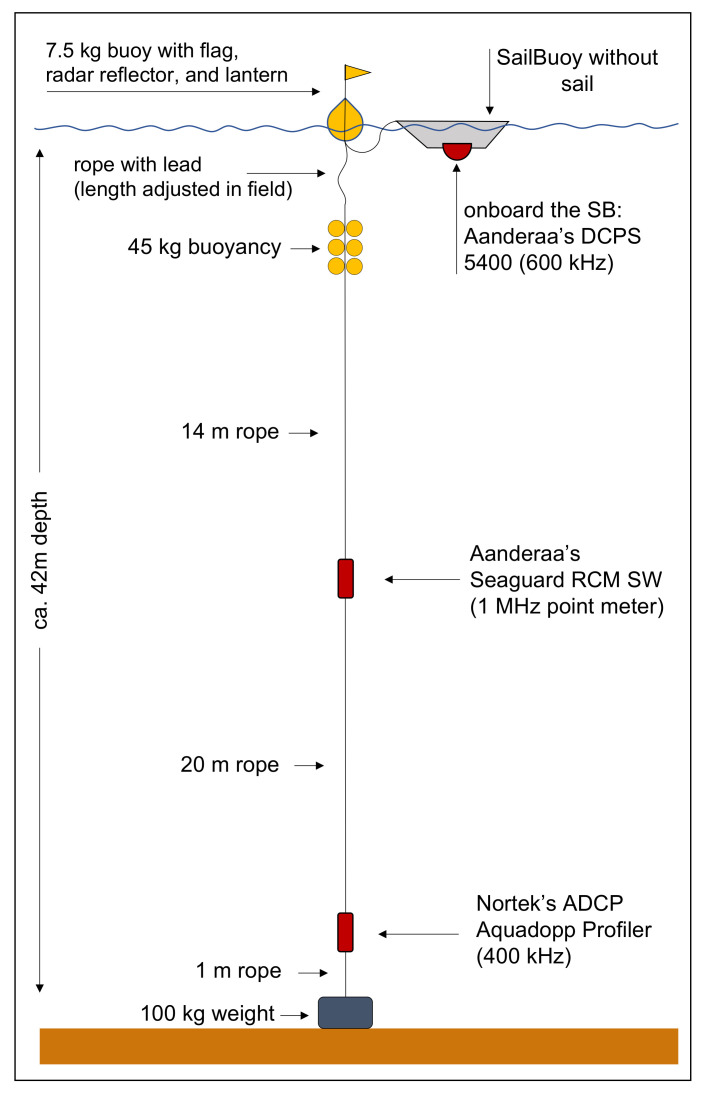
Schematic drawing of the instrumental setup of the stationary validation experiment in Fusafjorden in 2020.

**Figure 3 sensors-22-05553-f003:**
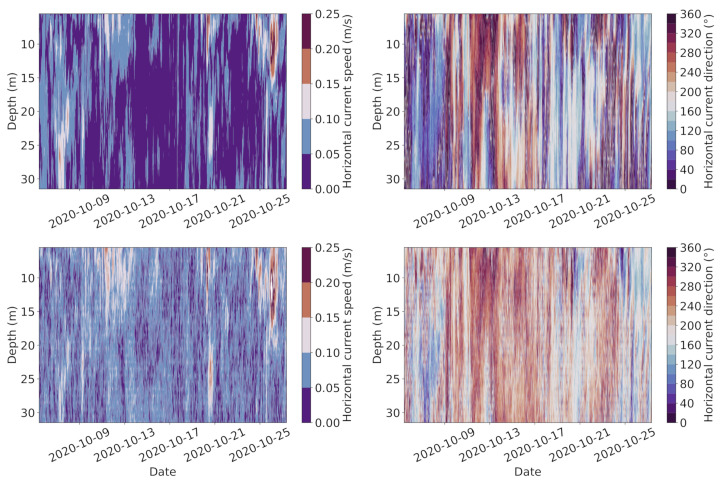
Horizontal speed (**left**) and current direction (**right**) measured by the SailBuoy ADCP (**top**) and the bottom-mounted ADCP (**bottom**), respectively, over time in Fusafjorden, October 2020.

**Figure 4 sensors-22-05553-f004:**
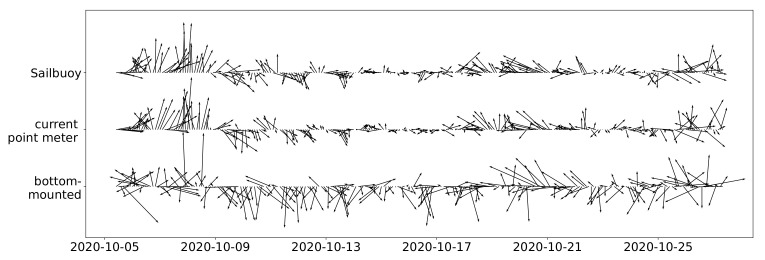
Current speed and direction measured by all three instruments at ca. 20 m depth over time in Fusafjorden, October 2020.

**Figure 5 sensors-22-05553-f005:**
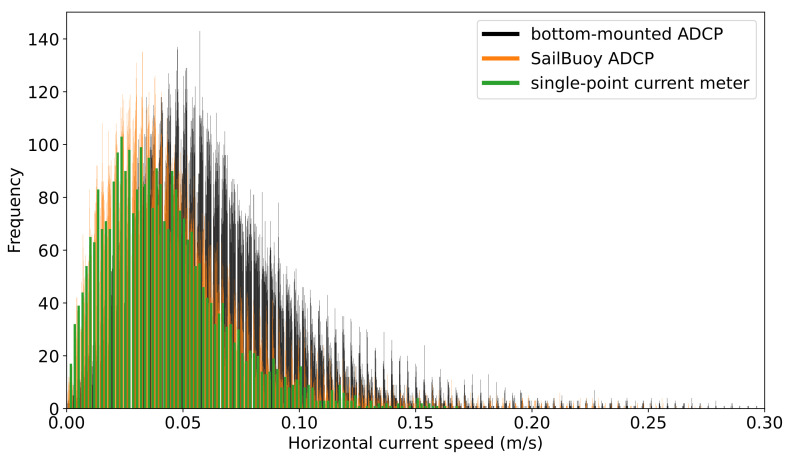
Histogram of the current speed data measured by the SailBuoy and bottom-mounted ADCPs and single-point current meter.

**Figure 6 sensors-22-05553-f006:**
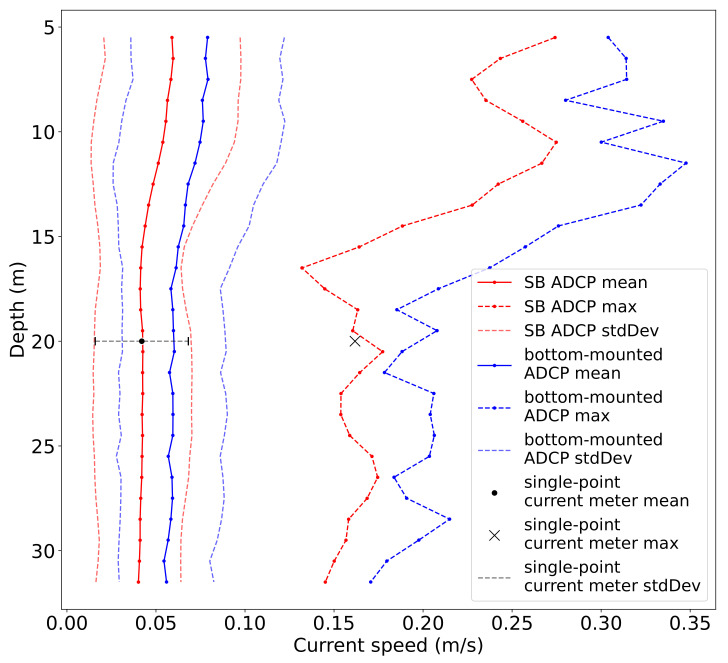
Time-averaged horizontal current speed and maximum speed over depth measured by the SB ADCP (blue), the bottom-mounted ADCP (red), and the single-point current meter (black) in Fusafjorden, October 2020. Standard deviations for the mean speed are displayed for reference.

**Figure 7 sensors-22-05553-f007:**
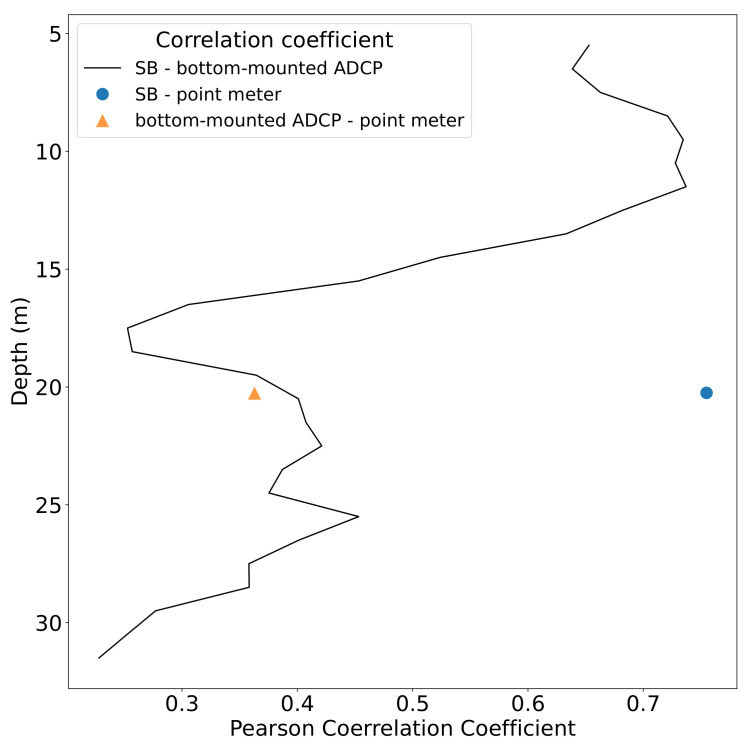
Depth profile of the Pearson correlation coefficient between the bottom-mounted and the SB profilers as well as the single-point current meter during stationary deployment in Fusafjorden.

**Figure 8 sensors-22-05553-f008:**
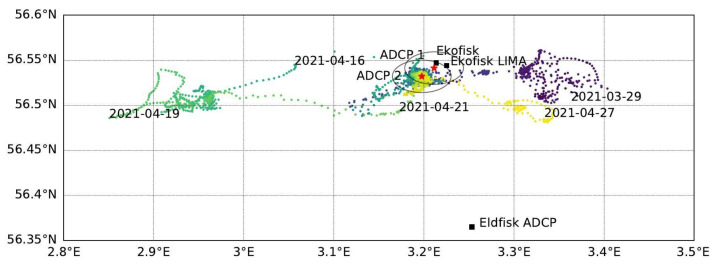
SB location during the Ekofisk measurement period in April 2021. The time is color-coded (starting at the end of March in purple, turning blue and green with time, and ending in yellow by the end of April). The locations of the moored ADCPs (red stars) and the Ekofisk platforms are marked for reference. Grey circles denote an approx. 2 km radius around the ADCPs.

**Figure 9 sensors-22-05553-f009:**
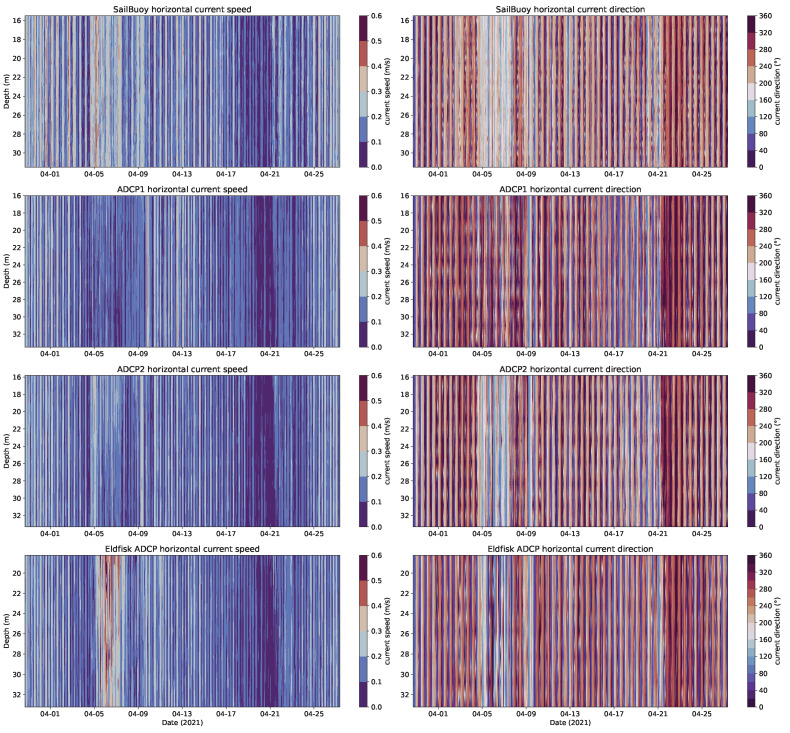
Current speed (**left**) and direction (**right**) measurements of the four profilers (SB, ADCP1 and 2, and Eldfisk) during the Ekofisk validation campaign in April 2021 showing results from approx. the same depth range.

**Figure 10 sensors-22-05553-f010:**
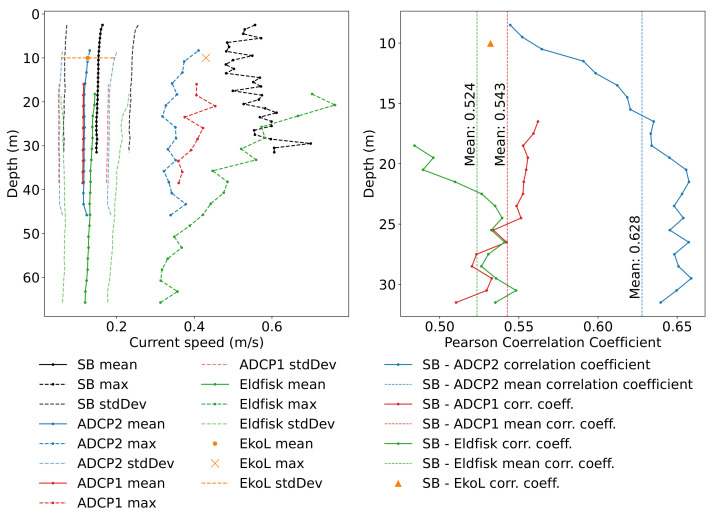
Ekofisk results. (**Left**): Comparison of mean and max speed of SB, ADCP1, ADCP2, Eldfisk, and the current meter at the Ekofisk Lima platform (Eko-L) over depth for the entire Ekofisk deployment period. Solid lines represent the mean current speed with its standard deviation given by lightly dashed lines around each curve. Dashed lines with markers represent the maximum speeds. (**Right**): correlation coefficients of SB with the ADCP1, ADCP2, Eldfisk profilers, and the Eko-L current meter (at 10 m depth), respectively.

**Figure 11 sensors-22-05553-f011:**
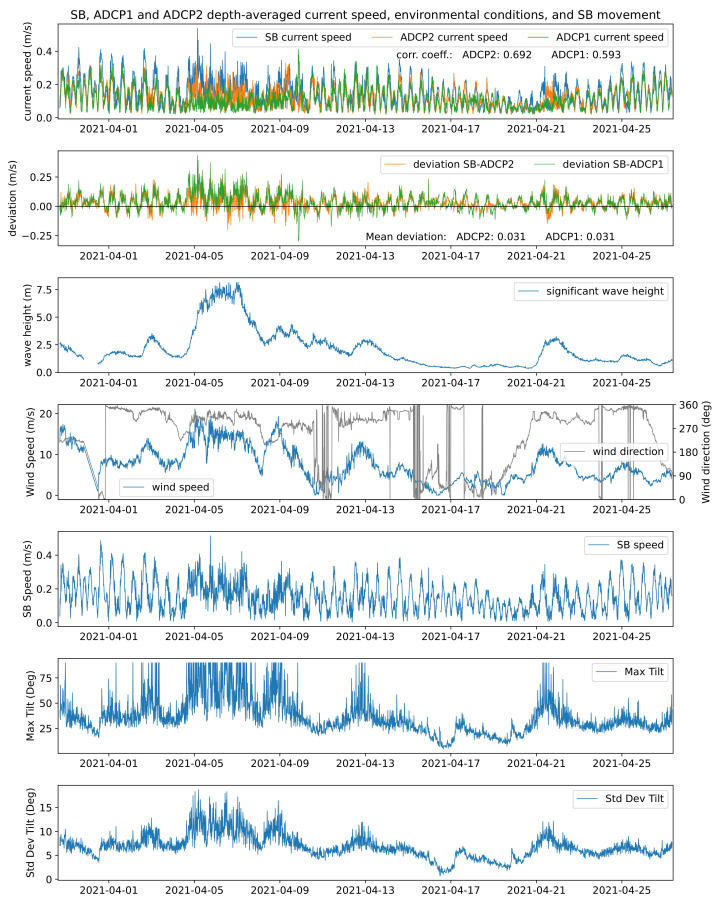
Overview of current speed and environmental conditions. The uppermost panel shows depth-averaged current speeds measured by the SailBuoy, ADCP1, and ADCP2, respectively, with the deviation between the datasets in the panel below. Environmental conditions (i.e., significant wave height and wind speed, wind (coming-from) direction, measured at the Ekofisk Lima platform (source: MET Norway), are displayed in panels three and four. The lower three panels represent data from the SailBuoy’s GPS and IMU, including the speed over ground of the SailBuoy as well as its maximum tilt and standard deviation of the tilt.

**Figure 12 sensors-22-05553-f012:**
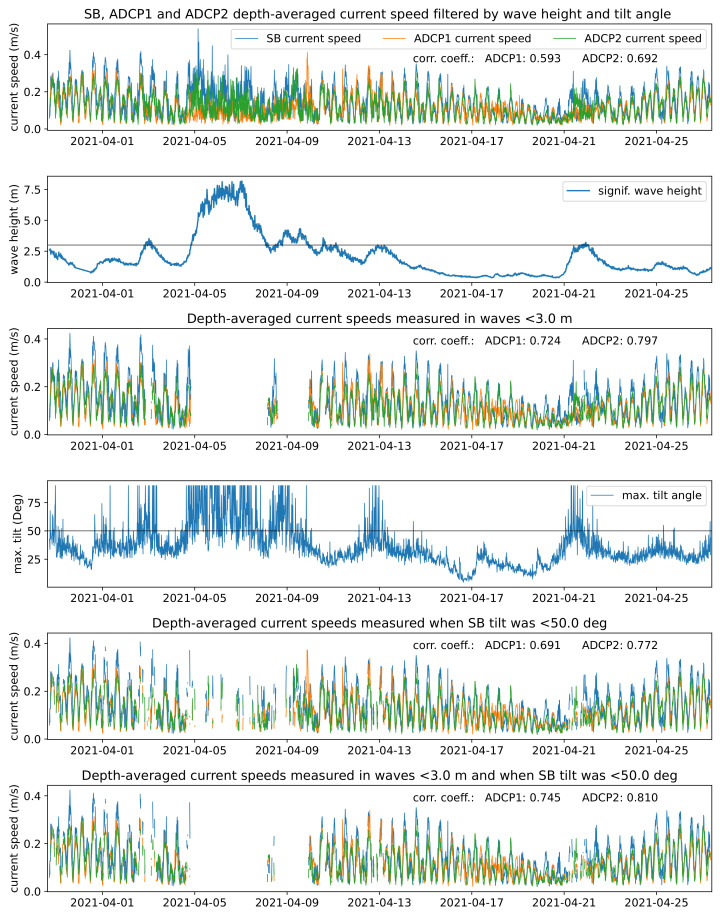
Ekofisk timeseries of current speeds measured by the SB filtered by wave height 3 m (3rd panel) and tilt angles 50° (5th panel) and both (bottom panel). Correlation coefficients of the SB with ADCP1 and ADCP2 data are annotated in the plots.

**Table 1 sensors-22-05553-t001:** Overview and instrument specifications of the different current measurement devices used in this study. ADCPs are profiling instruments, while the DCS (Doppler current sensor) is a point meter. The two numbers of pings per sampling interval given for the SB ADCP correspond to the Fusafjorden and the Ekofisk campaign, respectively.

Platform	Instrument Type and Model	Operating Frequency	Sampling Interval	Bin Size	Number of Pings
SailBuoy	ADCP Aanderaa DCPS 5400	600 kHz	15 min	2 m	750/150
Fusafjord mooring	DCS Aanderaa SeaGuard RCM SW	1 MHz	10 min	-	600
Fusafjord bottom- mounted	ADCP Nortek Aquadopp profiler	400 kHz	10 min	1 m	600
Ekofisk mooring (2×)	ADCP Nortek Aquadopp profiler	400 kHz	10 min	2.5 m	180
Eldfisk mooring	ADCP Nortek Aquadopp profiler	400 kHz	20 min	2.5 m	180
Ekofisk Lima mooring	Single-point current meter Nortek Aquadopp current meter	2 MHz	10 min	-	-

## Data Availability

All data presented are available on request. The Python scripts are available on request.

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
