# Peer review of "SailBuoy Ocean Currents: Low-Cost Upper-Layer Ocean Current Measurements"

_sensors, 2022, doi:10.3390/s22155553_

Round 1

Reviewer 1 Report

This paper presents an evaluation of current measurements made by an ADCP mounted on an unmanned sailing vessel.  The research design and evaluation of the data are very good, and the paper needs little revision for publication.  The results seem to show that the vessel mounted ADCP works well, but they also highlight the difficulty of finding "ground truth" current measurements.  It is particularly disturbing that the bottom mounted ADCP in the fjord test disagrees with point current meter.  The authors certainly give the bottom mounted ADCP the benefit of the doubt - I would be tempted to say that it is simply noisy.  Is there some way of checking that at this stage?  I think it would be very instructive to calculate the correlations between the various Ekofisk and Eldfisk current meters.  My suspicion is that those correlations are not much higher than those to the vessel mounded ADCP.  Finally, I don't like the shading in Figure 5.  It would be much easier to see the data without the shading.

Author Response

We would like to thank the reviewers for the positive feedback and the well informed review with good suggestions for improvements of our manuscript. In the following please find our answers to the suggestions, questions and comments. 

In response to reviewer #1:

  • Concerning the possibility to check whether the data of the bottom mounted ADCP is simply noisy: This point is certainly right, it is difficult to find the “ground truth” in current measurements, which complicates validation efforts. Checking the level of noise retrospectively is unfortunately not possible and only assumptions can be made. But the next suggestion by reviewer #1 might give a hint on the data quality.

  • Calculate correlations between the various reference instruments.

Thank you for this good suggestion. We have done so and included a statement into the results and discussion section of the manuscript. “For reference a correlation analysis was also calculated between the various reference instruments resulting in unsurprisingly high values of 0.70 for ADCP1 and ADCP2 and values around 0.57 and 0.46 for the correlation between Eldfisk and ADCP2 and ADCP1, respectively (not shown).” … “Furthermore, the correlation analyses applied to the datasets of the reference instruments resulting in correlation coefficients of similar magnitude as the SB ADCP data revealed the difficulty of obtaining accurate current measurements for validation.”Update shading in Figure 5: Figure has been updated.

Reviewer 2 Report

The paper presents an interesting addition to the senor package on board a sail buoy oceanographic drone. The results of ADCP measurements of ocean current for different sources are compared, and the authors claim an acceptable level of agreement between the results. 

Comparison of measurements of fluid velocity are extremely challenging, unless the fluid flow is uniform across the domain. For unsteady flow, the comparison will depend on the relative location of the measurement, the size of the area or volume used to calculate the flow vectors, as well as differences between instruments and how the instruments are placed in the environment. The Sail Buoy can expect to have extra challenges compared to a fixed device, as the platform itself is moving in all six degrees of freedom. Current calculations will need the combination of the ADCP measurements and the motion of the Sail Buoy, both in terms of drift and high frequency motions, which can change the path of the acoustic beam.  

Figure 8 shows that the Sail Buoy was a substantial distance away from the other measurement systems for quite a period of time. How confident are the authors that the current was uniform over the full domain? 

My big comment to the authors is can they reframe the results in terms of the uncertainties in the measurement systems? I agree that in general, more data is better, but we need to know what the uncertainty of all the measurements is when compared to a know reference current. 

I do not mean to diminish the work done by the authors, as mounting a significant sea trial with multiple data sources is a complex undertaking, and to complete it successfully is a great success. However, introducing more measurements, introduces more sources of uncertainty. My points are more for discussion than anything. 

Otherwise, I would suggest a couple of minor edits. 

Line 80: degrees of freedom is the accepted use...

Line 99: without the sail mounted on the vessel

Author Response

We would like to thank the reviewers for the positive feedback and the well informed review with good suggestions for improvements of our manuscript. In the following please find our answers to the suggestions, questions and comments.

  • On the question of confidence in the assumption of uniform current in the Ekofisk area.
    We would like to thank the reviewer for mentioning this point. This has been discussed thoroughly during data processing but might have come too short in the manuscript. We added a statement on this in the results: “Thorough data analysis, literature research (e.g., [13]) as well as weighing the amount of data against possible uncertainties due to small eddy features eventually led to the assumption of current conditions sufficiently homogeneous for the purpose of validation in the area of interest.”

  • Reframe results in terms of uncertainties in the measurement systems. We have tried to discuss the uncertainties better in the updated manuscript. Rev. 1 had a similar comment, and we looked at correlations between the various reference instruments and included a statement into the results and discussion section of the manuscript. “For reference a correlation analysis was also calculated between the various reference instruments resulting in unsurprisingly high values of 0.70 for ADCP1 and ADCP2 and values around 0.57 and 0.46 for the correlation between Eldfisk and ADCP2 and ADCP1, respectively (not shown).” … “Furthermore, the correlation analyses applied to the datasets of the reference instruments resulting in correlation coefficients of similar magnitude as the SB ADCP data revealed the difficulty of obtaining accurate current measurements for validation.”
  • Line 80: changed as suggested to “degrees of freedom”
  • Line 99: We believe you meant line 88 where the suggested line can be found. Here we changed the prior phrasing to the suggested.